# Improving Fairness in Image Classification via Sketching

**Ruichen Yao**[1]*    **Ziteng Cui**[2]*    **Xiaoxiao Li**[1]†    **Lin Gu**[3]†

[1]Department of Electrical and Computer Engineering, University of British Columbia
[2]Department of Advanced Interdisciplinary Studies, The University of Tokyo
[3]Machine Intelligence for Medical Engineering Team, RIKEN AIP
[1]{yrc0224, xiaoxiao.li}@ece.ubc.ca
[2]cui@mi.t.u-tokyo.ac.jp
[3]lin.gu@riken.jp

## Abstract

Fairness is a fundamental requirement for trustworthy and human-centered Artificial Intelligence (AI) system. However, deep neural networks (DNNs) tend to make unfair predictions when the training data are collected from different sub-populations with different attributes (*i.e.* color, sex, age), leading to biased DNN predictions. We notice that such a troubling phenomenon is often caused by data itself, which means that bias information is encoded to the DNN along with the useful information (*i.e.* class information, semantic information). Therefore, we propose to use sketching to handle this phenomenon. Without losing the utility of data, we explore the image-to-sketching methods that can maintain useful semantic information for the target classification while filtering out the useless bias information. In addition, we design a fair loss to further improve the model fairness. We evaluate our method through extensive experiments on both general scene dataset and medical scene dataset. Our results show that the desired image-to-sketching method improves model fairness and achieves satisfactory results among state-of-the-art (SOTA).

## 1 Introduction

Neural networks make many great achievements in different computer vision scenarios including general scenes and medical scenes, which bring much wellness to society. However, serious challenge also ensues. Standard vision model tends to make unfair predictions on skin color, age, sex, *etc.* [1, 2, 10, 12, 14, 18, 22, 27, 28]. One key reason comes from the bias in the input data. When the images are fed into the network, the bias information is also encoded thus leading model's unfair prediction.

Taking advantage of the advances in the latest image-to-sketch research [4, 7, 8, 13, 24, 26], we exploit the sketch to relieve unfairness while keeping the class or semantic information. Since sketches convey most semantic information from the original images, neural networks could also use them to make predictions without sacrificing much accuracy, especially where class discrimination relies little on the color information (*i.e.* skin cancer). Meanwhile, sketches throw the bias factors (*i.e.* color, texture) out with the color and texture information.

Previous methods often handle the unfairness problem in three ways: Pre-processing, In-processing, and Post-processing. (a). The pre-processing work deals with the bias in the database [19, 21,

---

*The two authors contributed equally to this paper
†Corresponding authors

2022 Trustworthy and Socially Responsible Machine Learning (TSRML 2022) co-located with NeurIPS 2022.

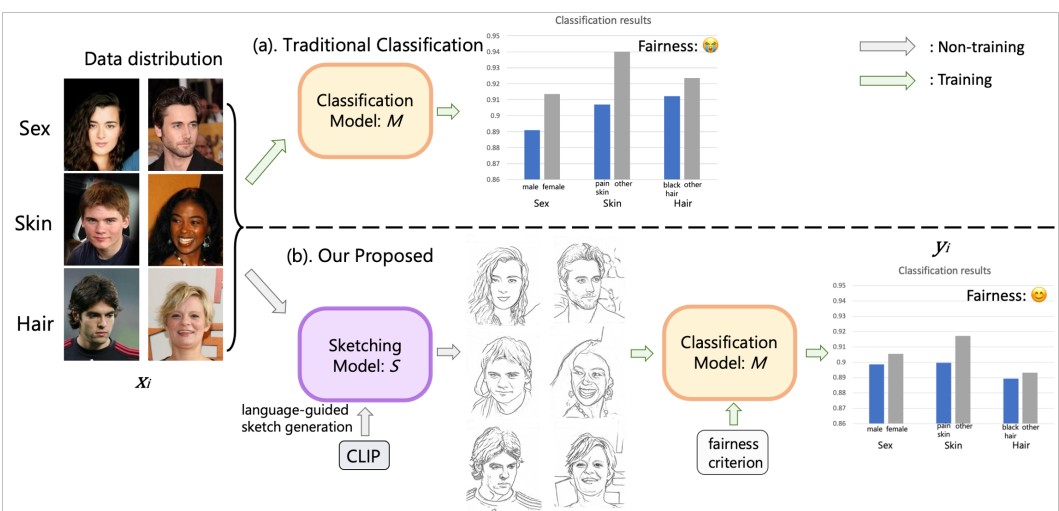

Figure 1: (a) is original classification pipeline. (b) is the overall pipeline of our proposed method, which we first send input image $x_i$ to sketching model $S$ and then train the sketch images $S(x_i)$ with additional fairness criterion. Note that the green arrow means the pipeline participates in training and the gray means the pipeline is offline in training.

27, 31], which tries to reduce the bias contained in the training set, such as transforming the data representation [19], generating paired training data to balance protected attributes [21], and generating adversarial examples to balance the number of protected attributes in the training set [31]. In this work, the way to process the input images into sketches is also pre-processing approach. (b). In-processing method aims to introduce a fairness mechanism in the training model [3, 14, 29, 30], such as adding fairness loss to constrain [29] and using adversarial training to force the model to produce a fair output [3, 14, 30]. The purpose is to obtain fairer models by explicitly changing the training procedure. (c). Post-processing approach adjusts model's predictions according to some fairness criteria [11, 16]. One way is to adjust the model predictions by editing the protected attributes accordingly [16], however this method is difficult to implement in practice [27]. Another work is to post-process a pre-trained deep learning model and create a new classifier with the same accuracy for people with different protected attributes [11].

Without manipulating model in training stage or adjusting prediction results, we use image sketching as an intuitive solution to complete the pre-process step on the data side. Our contribution could be summarized as follow: (1). We make the first attempt to exploit sketching for improving computer vision fairness. This simple solution could effectively abandon the bias of input data for fair prediction. (2). We introduce a fairness loss to further improve model's fairness results. Our method achieves satisfactory performance among state-of-the-art.

## 2 Methods

### 2.1 Overview

An overview of our method has been shown in Fig. 1. Based on standard vision classification paradigm, our pipeline mainly involves two modifications. First, we convert the input images into their sketches and feed them to the following classification model for prediction. Towards further fairness improvement, in the second step we introduce a fairness loss function to mitigate the bias in the model. Sufficient experiments and detailed explanations are shown in Sec. 3.

For each image index $i$, $x_i$ denotes the original input images, $y_i$ denotes the image label, and $z_i$ denotes the bias sensitive attribute. In our method, the input images $x_i$ would first be sent into sketch generation model $S$ to generate $x_i$'s corresponding sketch $S(x_i)$. Then the sketch image $S(x_i)$ would be sent into following classifier $M$ to generate classification results $\hat{y}_i = M(S(x_i))$. Classification loss $\mathcal{L}_{\text{cls}}$ would by calculating the cross-entropy (CE) loss between $\hat{y}_i$ and $y_i$, additionally we design a fairness loss $\mathcal{L}_{\text{fair}}$ (see Sec. 2.3) for further fairness improvement. Overall, the classification model

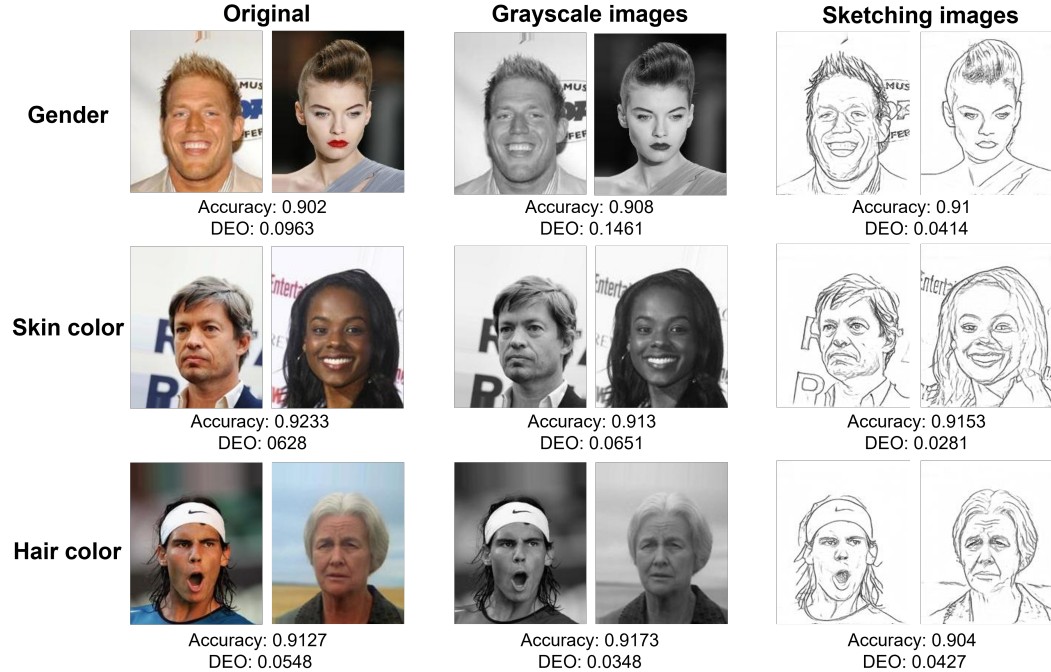

|  | Original | Grayscale images | Sketching images |
|---|---|---|---|

Figure 2: Experimental results of original photographs, grayscale images, and sketch images. Target label is smiling and sensitive attributes are gender, skin color, and hair color respectively.

$M$ would be optimized by the total loss $\mathcal{L}$ ($\lambda$ is a non-negative number to balance, which we set to 1.0 in our experiments):

$$\mathcal{L} = \mathcal{L}_{\text{cls}} + \lambda \cdot \mathcal{L}_{\text{fair}}. \tag{1}$$

As to evaluation metrics, based on previous works [14, 27], we apply *Statistical Parity Difference* (*SPD*), *Equal Opportunity Difference* (*EOD*), *Difference in Equalized Odds* (*DEO*), and *Average Odds Difference* (*AOD*) to measure and evaluate model's fairness. For all four evaluation metrics, the smaller the value of their indicators, the higher the fairness of the model.

## 2.2 Sketch Generation

For sketch generation model $S$, we employ the latest image-to-sketching method [4] to convert input image $x_i$ into its corresponding sketch $S(x_i)$.

The sketching method [4] is an unsupervised model trained on unpaired data between photographs (domain $A$) and sketches (domain $B$). It setups an adversarial training network with generators $G_A$, $G_B$ and discriminators $D_A$, $D_B$ respectively for domain $A$ and $B$. Different from previous image-to-sketching methods, this model not only focuses on image appearance but also explicitly instills geometric and semantic information into drawings. Specifically, the model is forced to draw lines in important geometric areas under a geometric loss which predicts depth information. In addition, a CLIP [20] model guidance is adopted to constrain the semantic meaning of sketches close to that of in original images [4, 20].

In the training stage, geometric loss $\mathcal{L}_{\text{Geom}}$ is implemented to maximize the depth information and capture as much depth information as possible. Using a SOTA depth estimation network $F$ [17], a network $G_{Geom}$ is designed to predict the depth map of sketches during training. Given input $a$, $G_A$ is the sketch generation network and $I(G(a))$ are the extracted ImageNet [6] features. The depth maps of photographs can be denoted as $F(a)$ and depth maps of sketches is $G_{Geom}(I(G(a)))$. The geometric loss is the absolute difference between these two predicted depth maps. $\mathcal{L}_{\text{Geom}} = |G_{Geom}(I(G(a))) - F(a)|$

The semantic loss encourages minimizing the semantic meaning gap between input images and line drawings. The smaller the semantic distance, the more accurate line drawings are translated into. The semantic loss is defined as the absolute CLIP difference between input images and sketches.
$\mathcal{L}_{\text{CLIP}} = |\text{CLIP}(G_A(a)) - \text{CLIP}(a)|$

Moreover, appearance loss $\mathcal{L}_{\text{cycle}}$ and adversarial loss $\mathcal{L}_{\text{GAN}}$ are implemented to encode input appearance and generate images belonging to their domains. The overall loss function is the weighted summation of these four losses:

$$\mathcal{L}_{\text{sketch}} = \lambda_{\text{CLIP}}\mathcal{L}_{\text{CLIP}} + \lambda_{\text{Geom}}\mathcal{L}_{\text{Geom}} + \lambda_{\text{GAN}}\mathcal{L}_{\text{GAN}} + \lambda_{\text{cycle}}\mathcal{L}_{\text{cycle}}. \tag{2}$$

Chan et al. [4] set $\lambda_{\text{CLIP}} = 10$, $\lambda_{\text{Geom}} = 10$, $\lambda_{\text{GAN}} = 1$, and $\lambda_{\text{cycle}} = 0.1$ in practice. Taking advantage of sketching, we are able to reduce the bias. As shown in Fig. 2, the sketch images are not inferior in accuracy and have an obvious advantage in the fairness indicator *DEO* when compared with the original images and the grayscale images.

## 2.3 Fairness Constraint

Since the initial purpose of image-to-sketching method [4, 8, 13, 24, 26] is not designed for fairness improvement, we introduce an additional fairness loss function to further mitigate bias of the model during training. The fairness loss function $\mathcal{L}_{\text{fair}}$ is designed as a mean square error (*MSE*) loss of *Statistical Parity Difference (SPD)* for every mini-batch. During model training, the fairness loss $\mathcal{L}_{\text{fair}}$ would combine with classification loss $\mathcal{L}_{\text{cls}}$ to optimize model parameters (see Eq. 1).

$$\mathcal{L}_{\text{fair}} = \frac{1}{n}\sum_{m=1}^{n}(SPD_{predict_m} - SPD_{ideal_m})^2 \tag{3}$$

Specifically, *SPD* measures the difference of probability in positive predicted label ($\hat{y} = 1$) between protected ($z = 1$) and unprotected ($z = 0$) attribute groups.

$$SPD = |P(\hat{y} = 1|z = 1) - P(\hat{y} = 1|z = 0)| \tag{4}$$

# 3 Experiments

## 3.1 Datasets and Experimental Setup

**Datasets.** We adopt two datasets for our evaluation, CelebA [15] and Skin ISIC 2018 [5, 25], CelebA for face recognition classification and Skin ISIC 2018 for skin cancer classification.

- CelebA
  The CelebA dataset consists of 202,599 human face images with 40 features per image [15]. Following Wang et al. [27], we choose smiling and attractive as the predicted label for binary class classification. For each predicted label, we choose gender (male and female), hair color (black hair and others), and skin color (painful skin and others) as the bias sensitive attributes. We select 10,000 images for every experiment, of which 70% for training, 15% for validation, and 15% for testing. For each experiment, the number of images in protected attribute and unprotected attribute are equal, which means we adopt a fair dataset.

- Skin ISIC 2018
  The Skin ISIC 2018 dataset consists of 10,015 skin images for 7-class skin cancer classification [5, 25]. Following Li et al. [14], we choose gender (male and female), age ($\geq 60$ and $<$ 60), and skin color (dark and light skin) as the bias sensitive attributes. We randomly split the dataset into training set (80%) and testing set (20%).

**Training and Evaluation.** We use original images and sketch images to train the classification model separately. Furthermore, we compare with grayscale images whose color information is removed. This allows us to evaluate the impact between changing only the color of the image and modifying both the color and texture. In addition, the fairness loss function is added to cross-entropy loss when training on sketch images (see Eq. 1).

We use ResNet-18 [9] to train CelebA dataset and VGG11 [23] to train ISIC dataset following previous setting [14, 27]. Batch size is set to 64 in CelebA and 32 in ISIC. Learning rate is set to

1e-3. For evaluation metrics, we evaluate the results CelebA dataset with *SPD* and *DEO*. And we evaluate the results of ISIC dataset with *SPD*, *EOD*, and *AOD*.

## 3.2 Experimental Results

On CelebA dataset, tables from 1 to 6 show that classification accuracy does not drop too much on other styles of images compared to the one on original images. Particularly, sketches either with or without fairness loss exhibit the minimum values of fairness metrics, *SPD* and *DEO*, showing the highest fairness compared to the original and grayscale images. The result of sketches with $\mathcal{L}_{\text{fair}}$ demonstrates that the fairness loss could further improve fairness in some cases.

Table 1: Results on CelebA when the target label is smiling and sensitive attribute is gender

|  | ACC ↑ | SPD ↓ | DEO ↓ |
|---|---|---|---|
| Original | 0.902 | 0.14 | 0.0963 |
| Grayscale | 0.908 | 0.1787 | 0.1461 |
| Fair training+FAAP | **0.9249** | 0.1326 | **0.0281** |
| **Sketches w/o $\mathcal{L}_{\text{fair}}$** | 0.91 | 0.1107 | 0.0414 |
| **Sketches w/ $\mathcal{L}_{\text{fair}}$** | 0.9087 | **0.1053** | 0.0682 |

Table 2: Results on CelebA when the target label is attractive and sensitive attribute is gender

|  | ACC ↑ | SPD ↓ | DEO ↓ |
|---|---|---|---|
| Original | **0.8127** | 0.5067 | 0.5864 |
| Grayscale | 0.8167 | 0.4827 | 0.5232 |
| Fair training+FAAP | 0.7931 | **0.2244** | **0.0434** |
| **Sketches w/o $\mathcal{L}_{\text{fair}}$** | 0.7893 | 0.4253 | 0.4457 |
| **Sketches w/ $\mathcal{L}_{\text{fair}}$** | 0.7007 | 0.2987 | 0.2811 |

Table 3: Results on CelebA when the target label is smiling and sensitive attribute is skin color

|  | ACC ↑ | SPD ↓ | DEO ↓ |
|---|---|---|---|
| Original | **0.9233** | 0.1613 | 0.0628 |
| Grayscale | 0.916 | 0.1307 | 0.0651 |
| **Sketches w/o $\mathcal{L}_{\text{fair}}$** | 0.9153 | 0.132 | **0.0281** |
| **Sketches w/ $\mathcal{L}_{\text{fair}}$** | 0.908 | **0.1040** | 0.0495 |

Table 4: Results on CelebA when the target label is attractive and sensitive attribute is skin color

|  | ACC ↑ | SPD ↓ | DEO ↓ |
|---|---|---|---|
| Original | **0.8093** | 0.2787 | 0.3371 |
| Grayscale | 0.8027 | 0.3107 | 0.3562 |
| **Sketches w/o $\mathcal{L}_{\text{fair}}$** | 0.7807 | **0.1947** | 0.204 |
| **Sketches w/ $\mathcal{L}_{\text{fair}}$** | 0.78 | 0.196 | **0.1409** |

Table 5: Results on CelebA when the target label is smiling and sensitive attribute is hair color

|  | ACC ↑ | SPD ↓ | DEO ↓ |
|---|---|---|---|
| Original | 0.9127 | 0.0613 | 0.0548 |
| Grayscale | **0.9173** | 0.052 | 0.0348 |
| **Sketches w/o $\mathcal{L}_{\text{fair}}$** | 0.904 | 0.0547 | 0.0427 |
| **Sketches w/ $\mathcal{L}_{\text{fair}}$** | 0.8913 | **0.0373** | **0.0122** |

Table 6: Results on CelebA when the target label is attractive and sensitive attribute is hair color

|  | ACC ↑ | SPD ↓ | DEO ↓ |
|---|---|---|---|
| Original | **0.802** | 0.0707 | 0.0916 |
| Grayscale | 0.7913 | 0.0467 | 0.1412 |
| **Sketches w/o $\mathcal{L}_{\text{fair}}$** | 0.7793 | **0.0227** | 0.0848 |
| **Sketches w/ $\mathcal{L}_{\text{fair}}$** | 0.7767 | 0.0333 | **0.0564** |

We also compare a SOTA fairness method, fair training + FAAP [27]. As shown in table 1, our model outperforms [27] in terms of *SPD* on gender attribute. For accuracy and *DEO*, our model still delivers competitive performance.

While Wang et al. [27] only consider gender as sensitive attribute, we also assess other sensitive attributes that are likely to cause bias such as skin color and hair color. Table 3 and 4 report the result on skin color as sensitive attribute. The accuracy on sketches doesn't drop much but the fairness indices are far below that of in original photographs, significantly reducing the bias in original images. Table 5 and 6 also show similar patterns. On both skin color and hair color, the use of sketch images is demonstrated to effectively improve fairness without much sacrifice in classification accuracy.

Table 7: Results on ISIC when the sensitive attribute is gender

|  | Sex | Precision ↑ | Recall ↑ | F1-score ↑ | SPD ↓ | EOD ↓ | AOD ↓ |
|---|---|---|---|---|---|---|---|
| Original | Male | 0.847 | 0.826 | 0.828 | 0.114 | 0.143 | 0.17 |
|  | Female | 0.877 | 0.867 | 0.867 |  |  |  |
| Ours sketches w/o $\mathcal{L}_{\text{fair}}$ | Male | 0.81 | 0.741 | 0.769 | 0.136 | 0.131 | 0.161 |
|  | Female | 0.815 | 0.773 | 0.787 |  |  |  |
| Ours sketches w/ $\mathcal{L}_{\text{fair}}$ | Male | 0.835 | 0.779 | 0.799 | 0.12 | 0.125 | 0.152 |
|  | Female | 0.837 | 0.79 | 0.802 |  |  |  |

Table 8: Results on ISIC when the sensitive attribute is age

| | Age | Precision ↑ | Recall ↑ | F1-score ↑ | SPD ↓ | EOD ↓ | AOD ↓ |
|---|---|---|---|---|---|---|---|
| Original | ≥ 60 | 0.71 | 0.689 | 0.68 | 0.246 | 0.15 | 0.174 |
| | < 60 | 0.932 | 0.913 | 0.921 | | | |
| Ours sketches w/o $\mathcal{L}_{\mathrm{fair}}$ | ≥ 60 | 0.65 | 0.653 | 0.646 | 0.191 | 0.111 | 0.149 |
| | < 60 | 0.896 | 0.801 | 0.839 | | | |
| Ours sketches w/ $\mathcal{L}_{\mathrm{fair}}$ | ≥ 60 | 0.647 | 0.592 | 0.612 | 0.223 | 0.136 | 0.148 |
| | < 60 | 0.904 | 0.74 | 0.804 | | | |

Table 9: Results on ISIC when the sensitive attribute is skin tone

| | Skin tone | Precision ↑ | Recall ↑ | F1-score ↑ | SPD ↓ | EOD ↓ | AOD ↓ |
|---|---|---|---|---|---|---|---|
| Original | Dark | 0.853 | 0.803 | 0.822 | 0.134 | 0.201 | 0.203 |
| | Light | 0.935 | 0.932 | 0.932 | | | |
| Ours sketches w/o $\mathcal{L}_{\mathrm{fair}}$ | Dark | 0.782 | 0.72 | 0.742 | 0.134 | 0.179 | 0.181 |
| | Light | 0.847 | 0.796 | 0.818 | | | |
| Ours sketches w/ $\mathcal{L}_{\mathrm{fair}}$ | Dark | 0.779 | 0.698 | 0.714 | 0.144 | 0.133 | 0.155 |
| | Light | 0.843 | 0.808 | 0.82 | | | |

Tables 7 to 9 are the experimental results of ISIC dataset. In table 7, the sensitive attribute is gender. We find that although using sketches does not show more fairness in terms of *SPD*, the gap between the original images and the sketch images is very small. In terms of *EOD* and *AOD*, using sketch images can still effectively reduce bias. In addition, the decline of bias attributes is more significant when fairness loss is added. In table 8 and 9, we can find similar patterns as table 7. After adding fairness loss to the model, *EOD* and *AOD* decrease significantly.

# 4   Conclusion and Limitation

In this paper, we propose to use image sketching methods to improve model fairness among several bias types (*i.e.* color, sex, age) based on the intuition that a suitable sketching method can filter out the bias information while keeping semantic information for classification. We employ the latest image sketching method [4] with CLIP's [20] language encoder guidance to ensure that the images retain category and semantic information. Moreover, we also introduce an additional fairness loss to improve model's fairness further. We have evaluated our method, showing that image sketching is a simple but effective way to improve fairness.

We note that our method may cause accuracy drop, especially when we add the fairness loss $\mathcal{L}_{\mathrm{fair}}$ as additional supervision. This may be because that bluntly adding fairness makes the model more difficult to converge. In the future study, we're going to improve optimization strategies and combine this improvement with some sketch adjustment methods (*i.e.* style transfer), which may tackle the accuracy drop issue and even effectively improve the model fairness among all bias settings.

# Acknowledgments

This work is supported in part by the Natural Sciences and Engineering Research Council of Canada (NSERC) DGECR-2022-00430, the University of British Columbia's Health Innovation Funding Investment (HIFI) Awards, JST Moonshot R&D Grant Number JPMJMS2011, and JST ACT-X Grant Number JPMJAX190D, Japan.

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

# A    Fairness evaluation metrics

Let $x_i, y_i, z_i$ as the original input images, label, and bias sensitive attribute for every image $i$ in the dataset. $S(x_i)$ can be represented as sketch image and $M(S(x_i))$ is the predicted label $\hat{y}_i$. The true positive rate (*TPR*) and false positive rate (*FPR*) are:

$$TPR_z = P(\hat{y}_i = y_i | z_i = z) \tag{5}$$

$$FPR_z = P(\hat{y}_i \neq y_i | z_i = z) \tag{6}$$

Based on [14, 27], *Statistical Parity Difference (SPD)*, *Equal Opportunity Difference (EOD)*, *Difference in Equalized Odds (DEO)*, and *Average Odds Difference (AOD)* are applied to measure and evaluate the fairness. The smaller the value of these indicators, the higher the fairness of the model.

- *Statistical Parity Difference (SPD)* measures the difference of probability in positive predicted label ($\hat{y} = 1$) between protected ($z = 1$) and unprotected ($z = 0$) attribute groups.

$$SPD = |P(\hat{y} = 1 | z = 1) - P(\hat{y} = 1 | z = 0)| \tag{7}$$

- *Equal Opportunity Difference (EOD)* measures the difference of probability in positive predicted label ($\hat{y} = 1$) between protected ($z = 1$) and unprotected ($z = 0$) attribute groups given positive target labels ($y = 1$). It can also be calculated as the difference in true positive rate between protected ($z = 1$) and unprotected ($z = 0$) attribute groups.

$$EOD = |TPR_{z=1} - TPR_{z=0}| = |P(\hat{y} = 1 | y = 1, z = 1) - P(\hat{y} = 1 | y = 1, z = 0)| \tag{8}$$

- *Difference in Equalized Odds (DEO)* measures the difference of probability in positive predicted label ($\hat{y} = 1$) between protected ($z = 1$) and unprotected ($z = 0$) attribute groups given target labels ($y \in \{0, 1\}$).

$$DEO = |P(\hat{y} = 1 | y, z = 1) - P(\hat{y} = 1 | y, z = 0)|, y \in \{0, 1\} \tag{9}$$

- *Average Odds Difference (AOD)* measures the average of difference in true positive rate (*TPR*) and false negative rate (*FPR*) between protected ($z = 1$) and unprotected ($z = 0$) attribute groups.

$$AOD = \frac{1}{2}(|TPR_{z=1} - TPR_{z=0}| - |FPR_{z=1} - FPR_{z=0}|) \tag{10}$$

