# OpenReview forum: "Improving Fairness in Image Classification via Sketching"
_NeurIPS.cc/2022/Workshop/TSRML — TSRML2022_

### Official Review · Reviewer_tr5V · 2022-10-20
**Review of #133**

**Overall Rating:** 6

**Summary:**

This paper considers the fairness problem in computer vision. It proposes a pre-processing method to alleviate the bias in data. Specifically, it adopts an image-to-sketching method to convert each image to a sketch which can directly eliminate the bias caused by color. Then, it trains the model with sketches with a fairness loss. The proposed method shows a good performance in decreasing bias on two datasets with different settings of labels and sensitive attributes.

**Strengths:**

1. The paper is well-written and clear.
2. Although it uses an existing image-to-sketching method, the idea of using sketches in pre-processing is novel and it's good to see it works well in many settings.
3. The paper verifies the effectiveness of this method with plenty of experiments.

**Weaknesses:**

1. The limitation of using sketches is not well discussed. For example, it heavily relies on the quality of the sketching model, and it may not be a good solution in some scenarios. I'm concerned about the quality of skin image sketches for the ISIC dataset. And color might be an important factor in skin cancer diagnosis.
2. In experiments, the proposed method only compares with SOTA in two settings (Table 1 and 2) and it seems SOTA performs better. What's the performance of SOTA in other settings?
Other Questions:
1. What's the training data of the Sketch model? Are you using a pre-trained Sketch model? Is there an adaptation problem when you use a Sketch model trained from other data to sketch the CelebA or ISIC?
2. In experiments, the paper tests settings when the sensitive attribute is gender, race, and age. It makes sense that when using sketches the result becomes fairer if the sensitive attribute is race. What's the intuition behind when the sensitive attribute is gender? Why can sketches also mitigate bias?

**Overall Recommendation:**

Although this paper needs some effort to improve itself, the idea of using sketches to mitigate bias in computer vision is interesting.

**Review Confidence:**

3: The reviewer is fairly confident that the evaluation is correct

---

### Official Review · Reviewer_V68U · 2022-10-20
**The paper proposes and evaluates image-to-sketch as a methodology for reducing bias in classification models**

**Overall Rating:** 7

**Summary:**

The paper leverages recent research in image-to-sketch methods to obtain a sketch representation of an image. Since this image only contains crucial characteristics of an image, the work argues that it would throw away bias factors and thus result in fairer classification model. The paper also proposes a fairness loss and shows the applicability of the proposed methodology on two datasets and multiple fairness metrics.

**Strengths:**

- The idea of using sketch representation of an image for mitigating fairness concerns is novel.
- The paper performs detailed evaluation of the proposed methodology on two large scale datasets across multiple fairness metrics


**Weaknesses:**

- If the image-to-sketch model is itself biased - for instance it could create better sketches for certain combination of skin color and hair types, the whole methodology would fail. It would help to evaluate this limitation.

**Overall Recommendation:**

I propose for acceptance of this paper. Th methodology proposed is novel and the evaluation is thorough. However, I'd suggest evaluating how biasness in image-to-sketch model can propagate to the final outcome.

**Review Confidence:**

3: The reviewer is fairly confident that the evaluation is correct

---

### Official Review · Reviewer_uHmm · 2022-10-22

**Overall Recommendation:** I recommend acceptance.
**Overall Rating:** 7

**Summary:**

This paper studies how to improve fairness in image classification tasks by using image sketching. The authors propose a fair loss, combined with the image-to-sketching, the proposed method outperforms existing SOTA on certain benchmark datasets.

**Strengths:**

1. Experiments in this paper demonstrate the effectiveness of the proposed method on improving fairness (DEO, SPD) in image classification.

**Weaknesses:**

1. Compared to existing baselines, the proposed method slightly hurts the accuracy of the model on several datasets. This could be an interesting future direction.
2. The way this paper employs CLIP models is interesting, which could shed lights on future work.

**Review Confidence:**

4: The reviewer is confident but not absolutely certain that the evaluation is correct

---

### Decision · Program_Chairs · 2022-10-23

Accept